# Knowledge Distillation from Cross Teaching Teachers for Efficient Semi-Supervised Abdominal Organ Segmentation in CT

Jae Won Choi[1,2]

[1] Department of Radiology, Armed Forces Yangju Hospital, South Korea
[2] College of Medicine, Seoul National University, South Korea
jhoci@snu.ac.kr

**Abstract.** For more clinical applications of deep learning models for medical image segmentation, high demands on labeled data and computational resources must be addressed. This study proposes a coarse-to-fine framework with two teacher models and a student model that combines knowledge distillation and cross teaching, a consistency regularization based on pseudo-labels, for efficient semi-supervised learning. The proposed method is demonstrated on the abdominal multi-organ segmentation task in CT images under the MICCAI FLARE 2022 challenge, with mean Dice scores of 0.8429 and 0.8520 in the validation and test sets, respectively. The code is available at https://github.com/jwc-rad/MISLight.

**Keywords:** Knowledge distillation · semi-supervised learning · medical image segmentation.

## 1 Introduction

Organ segmentation has been one of the most popular applications of artificial intelligence in abdominal radiology [30]. As more high-quality imaging data are becoming available and advanced deep learning methods are being developed, many recent studies on automated abdominal organ segmentation have achieved promising results [16,2,8]. However, these methods are based on supervised learning that depends on large-scale, carefully labeled data. Also, current segmentation methods often require high computation costs. Therefore, for practical application in the clinical workflow, demands on labeled data and computational resources must be reduced.

Acquiring labeled data for medical image segmentation is especially expensive as it requires expert-level voxel-wise labeling and clinical data is innately heterogeneous. In this context, to utilize unlabeled data, various semi-supervised learning (SSL) in medical imaging have been studied, including Uncertainty-aware Mean Teacher [35], Uncertainty Rectified Pyramid Consistency [21], and Dual-task Consistency [19]. Among them, we adopt cross teaching, a simple consistency regularization based on pseudo-labels, which recently showed promising

results in semi-supervised medical image segmentation on cardiac MR data [20]. Also, we use models with slightly different decoders to boost the consistency regularization, following Mutual Consistency Training [34].

The main strategies to address the high computational cost of deep learning methods include (1) efficient building blocks and (2) model compression and acceleration techniques [7]. The latter has not gained as much interest as the former, especially in medical image segmentation [27], while there are many studies on lightweight networks [36,1]. Among model compression and acceleration techniques, knowledge distillation (KD), which refers to knowledge transfer from a larger teacher model to a smaller student model [11], has been applied increasingly in recent research [7,33]. The target knowledge to transfer can be the response of the last output layer, outputs of intermediate feature layers, or relationships between different feature maps [7]. Here, we apply the response-based KD because it is simple and can be implemented regardless of network architectures.

The current study proposes a coarse-to-fine framework (Figure 1) with two teacher models and a student model that combines KD and cross teaching, a consistency regularization based on pseudo-labels, for efficient semi-supervised medical image segmentation. Labeled data are used in all three models to train supervised segmentation. Pseudo-labels from unlabeled data are used to perform cross teaching between the two teachers and pseudo-supervision of the student. Meanwhile, outputs of the teachers on both labeled and unlabeled data are used to guide the student model through KD. Only the student model is used for efficient inference. The proposed method is developed and evaluated on the abdominal multi-organ segmentation task in CT images under the MICCAI FLARE 2022 challenge[1].

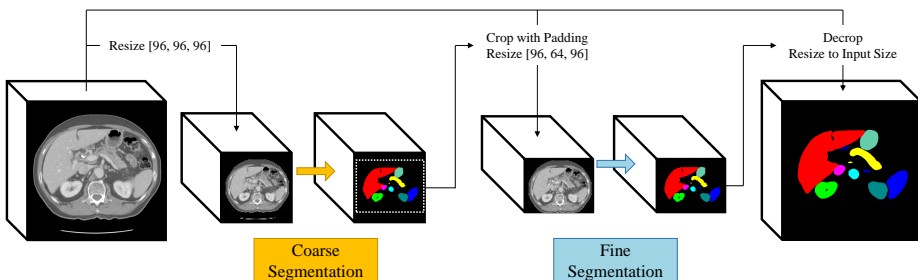

**Fig. 1.** An overview of the coarse-to-fine segmentation framework. For the coarse segmentation, the whole-volume input is resampled to $96 \times 96 \times 96$. For the fine segmentation, cropping with 10% padding around the coarse mask is first performed, and the cropped volume is resampled to $96 \times 64 \times 96$. The resultant fine segmentation mask is resized and padded back to the original input size.

---

[1] https://flare22.grand-challenge.org/

## 2   Method

### 2.1   Preprocessing

The following preprocessing steps are performed in all experiments:

- Reorienting images to the right-anterior-inferior (RAI) view.
- For coarse segmentation, whole-volume resampling to fixed size $96 \times 96 \times 96$ with trilinear interpolation. For fine segmentation, cropping with 10% padding around the coarse mask (ground truth, if present), then resampling to fixed size $96 \times 64 \times 96$ with trilinear interpolation.
- Clipping based on the Hounsfield units to [-300, 300].
- Patch-wise intensity normalization with z-score normalization based on the mean and standard deviation of the voxel values.

### 2.2   Proposed Method

The proposed method is a coarse-to-fine framework, where coarse segmentation is first yielded from whole-volume input and then refined by fine segmentation (Figure 1). Such a two-stage framework lowers computation costs, especially in terms of memory use and running time, compared to the sliding window approach, which is a more common solution in medical image segmentation [36,32]. Empirically, a single-stage segmentation led to poor segmentation results and long inference time in large field-of-view or whole-body CT images. Each stage of the proposed framework consists of two teacher models $T_1$ and $T_2$ and a smaller student model $S$ which are trained simultaneously (Figure 2). At inference, only the student model is used.

**Supervised Segmentation** Labeled data are used to train supervised segmentation for all models. Recently, compound losses have been suggested as the most robust losses for medical image segmentation tasks [22]. For model prediction $P$ and label $Y$, we apply the sum of Dice loss [26] and focal loss [17] as the supervised segmentation loss:

$$L_{seg} = Dice(P,Y) + Focal(P,Y)$$

**Cross Teaching and Pseudo-supervision** For SSL of the teacher models $T_1$ and $T_2$, we use the cross teaching strategy adopted from Cross Teaching between CNN and Transformer [20] and inspired by Cross Pseudo-supervision [4] and Mutual Consistency Training [34]. These methods all train two models with network-level perturbations that supervise each other with pseudo-labels to encourage consistent outputs on the same input. They differ in the perturbation targets (initialization [4], upsampling method for decoder [34], and learning paradigm [20]). Here, to distinguish using pseudo-labels for training between teacher models from using them to train the student model, we refer to the former as cross teaching and the latter as pseudo-supervision. With predictions of

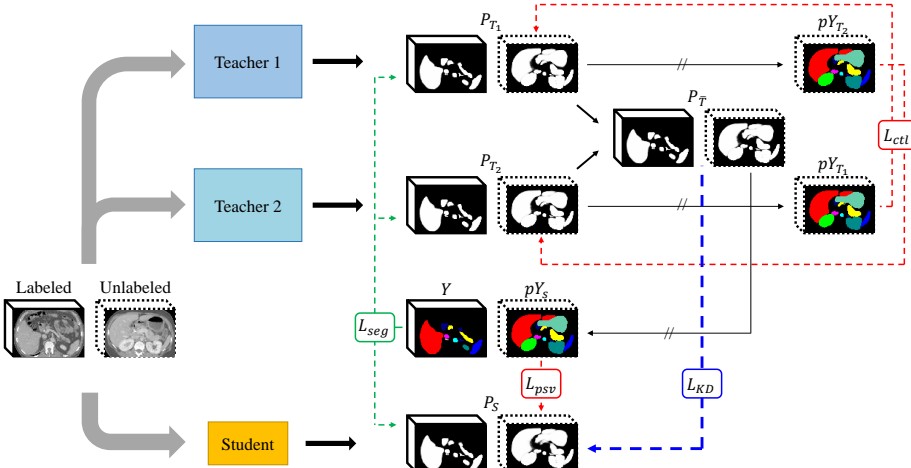

**Fig. 2.** An overview of KD from Cross Teaching Teachers. Each stage of the coarse-to-fine framework consists of two teacher models and a smaller student model. While labeled data are used in all three models to train supervised segmentation, the unlabeled data are used for cross teaching between the two teachers and pseudo-supervision of the student. All data are used for KD from the teacher models to the student model.

the student model $P_S$, teacher models $P_{T_1}$ and $P_{T_2}$, and teachers' mean $P_{\bar{T}}$, the cross teaching and pseudo-supervision losses for the unlabeled data are defined as:

$$L_{ctl} = Dice(P_{T_1}, argmax(P_{T_2})) + Dice(P_{T_2}, argmax(P_{T_1}))$$
$$L_{psv} = Dice(P_S, argmax(P_{\bar{T}}))$$

**Knowledge Distillation** The main idea of response-based KD is training the student model to directly mimic the final prediction of the teacher model. Following Hinton et al. [11], we apply the Kullback-Leibler (KL) divergence loss between $P_S$ and $P_{\bar{T}}$ on both labeled and unlabeled data. A weight factor $\lambda_{dis}$ is applied to balance distillation loss with the supervised segmentation loss for labeled data and the cross teaching and pseudo-supervision losses for unlabeled data:

$$L_{labeled} = L_{seg} + \lambda_{dis}KL(P_S, P_{\bar{T}})$$
$$L_{unlabeled} = L_{ctl} + L_{psv} + \lambda_{dis}KL(P_S, P_{\bar{T}})$$

Moreover, the proposed method is an online distillation where both the teachers and student models are updated simultaneously [7].

**Overall Objective** The overall training objective of the proposed method is the weighted sum of $L_{labeled}$ and $L_{unlabeled}$ with a weight factor $\lambda_{ssl}$ defined as:

$$Loss = L_{labeled} + \lambda_{ssl}L_{unlabeled}$$

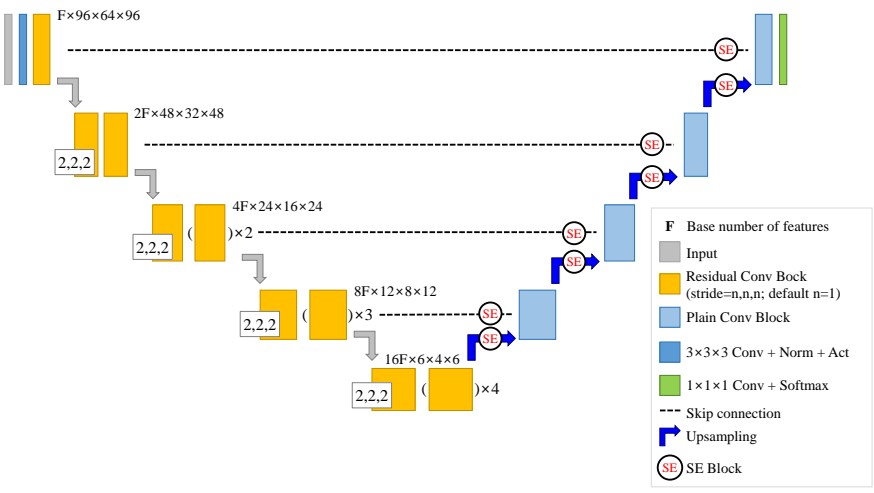

**Fig. 3.** Residual-USE-Net architecture. Mobile-Residual-USE-Net uses depthwise separable convolutions for residual and plain convolution blocks.

**Network Architecture** An overview of the network architecture is shown in Figure 3.

Inspired by the residual variant of the nnU-Net framework [14,15] and USE-Net [28], we employ Residual-USE-Net, a 3D U-Net [5] with an encoder with residual convolution blocks and a decoder with plain convolution blocks incorporated with residual squeeze-and-excitation (SE) blocks [13]. A convolution block is implemented as two sets of convolution, normalization, and nonlinear activation layers, and for the residual block, the residual summation takes place before the last activation. We set $r = 8$ for the reduction ratio of SE blocks [13,28] (Figure 4).

The teacher models $T_1$ and $T_2$ are Residual-USE-Nets with 32 base features and 4 skip connections. Following mutual consistency training [34], while $T_1$ and $T_2$ share the same encoder structure, their decoders use different upsampling methods where $T_1$ uses transposed convolutions and $T_2$ uses trilinear interpolation followed by regular convolutions.

The student and teacher models share the same overall network structure, but we apply depthwise separable convolutions as in MobileNets [12] to build a lighter neural network for the student model. The student model $S$ is Mobile-Residual-USE-Net, a Residual-USE-Net with depthwise separable convolutions

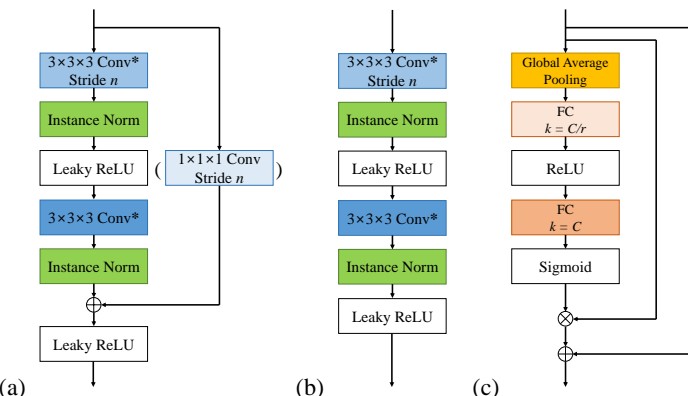

**Fig. 4.** (a) Residual convolution block. If stride $n = 1$ and the number of input and output channels are the same, the residual connection uses an identity layer instead of $1 \times 1 \times 1$ convolution. (b) Plain convolution block. (c) Residual SE block. *These layers are replaced with depthwise separable convolutions for Mobile-Residual-USE-Net.

instead of regular convolutions except for the initial convolution layer, with 32 base features, 4 skip connections, and transposed convolutions for the decoder.

### 2.3   Post-processing

The largest connected component of the segmentation mask is extracted per each class for both coarse and fine outputs. The connected component analysis is performed using Python connected-components-3d[2] and fastremap[3] packages [36].

## 3   Experiments

### 3.1   Dataset and evaluation measures

The MICCAI FLARE 2022 is an extension of the FLARE 2021 [23] with more segmentation targets and more diverse images. The dataset is curated from more than 20 medical groups under the license permission, including MSD [29], KiTS [9,10], AbdomenCT-1K [24], and TCIA [6]. The training set includes 50 labeled CT scans with pancreas disease and 2000 unlabeled CT scans with liver, kidney, spleen, or pancreas diseases. The validation set includes 50 CT scans with liver, kidney, spleen, or pancreas diseases. The testing set includes 200 CT scans where 100 cases has liver, kidney, spleen, or pancreas diseases and the other 100 cases has uterine corpus endometrial, urothelial bladder, stomach, sarcomas, or ovarian diseases. All the CT scans only have image information and the center information is not available.

---

[2] https://github.com/seung-lab/connected-components-3d
[3] https://github.com/seung-lab/fastremap

The evaluation measures consist of two accuracy measures: Dice Similarity Coefficient (DSC) and Normalized Surface Dice (NSD), and three running efficiency measures: running time, area under GPU memory-time curve, and area under CPU utilization-time curve. All measures will be used to compute the ranking. Moreover, the GPU memory consumption has a 2 GB tolerance.

### 3.2   Implementation details

**Environment settings**  The environments and requirements are presented in Table 1.

**Table 1.** Environments and requirements.

| | |
|---|---|
| Windows/Ubuntu version | Ubuntu 20.04 |
| CPU | AMD Ryzen Threadripper PRO 3975WX |
| RAM | 251G |
| GPU (number and type) | NVIDIA GeForce RTX 3090 (24G, ×1) |
| CUDA version | 11.4 |
| Programming language | Python 3.9 |
| Deep learning framework | PyTorch (torch 1.10.0, torchvision 0.11.1) |
| Code available at | https://github.com/jwc-rad/MISLight |

**Training protocols**  The training protocols are shown in Table 2. Except for the preprocessing, coarse and fine segmentation training are performed with the same protocols. During training, the labeled and unlabeled data are randomly sampled alternatively at a ratio of 1:1. An epoch is defined as an iteration over all the labeled data. Therefore, each epoch includes a random subset of the unlabeled data.

The weight factors $\lambda_{dis}$ and $\lambda_{ssl}$ are time-dependent Gaussian warming-up functions [35] $\lambda(t) = \lambda_0 \cdot e^{-5(1-t/t_{max})^2}$ where $t$ denotes the current training epoch and $t_{max}$ is the total epoch number. We use $\lambda_0 = 10$ for $\lambda_{dis}$ [18] and $\lambda_0 = 0.1$ for $\lambda_{ssl}$ [35].

The coarse segmentation is first trained using the whole-volume inputs. Then, the trained student model is applied to all the unlabeled data to acquire coarse masks. For the fine segmentation, cropping is performed around the coarse masks and the ground truth masks for the unlabeled and labeled data, respectively. Using the cropped volumes as inputs, the fine segmentation training is performed.

**Testing protocols**  Only the student is used at inference, with the number of model parameters 5.2M and the number of flops 21.7G.

---

[4] https://github.com/PyTorchLightning/pytorch-lightning

[5] https://github.com/sovrasov/flops-counter.pytorch

**Table 2.** Training protocols.

| | |
|---|---|
| Data augmentation | Elastic deformation, scaling, rotation, crop, Gaussian noise, brightness |
| Network initialization | Xavier normal initialization |
| Batch size | 1 |
| Patch size | $96 \times 64 \times 96$ |
| Total epochs | 1000 |
| Optimizer | SGD with nesterov momentum ($\mu = 0.99$, $decay = 3e - 5$) |
| Loss | Dice + Focal ($\alpha = 0.5$, $\gamma = 2$) |
| Initial learning rate | 0.01 |
| Learning rate decay schedule | $(1 - epoch/epoch_{max})^{0.9}$ [3] |
| Training time | 7.5 hours |
| Number of model parameters | 189.3M (5.2M in test) [4] |
| Number of flops | 443.1G (21.7G in test) [5] |

The same preprocessing as the training protocols except for data augmentation is applied for the testing. For coarse segmentation, inference is performed with a sliding window approach with overlap by half of the size of a patch where the resulting prediction is a weighted sum of sliding windows. To reduce the influence of predictions close to boundaries, a Gaussian importance weighting is applied for each predicted patch [14]. For fine segmentation, since the image is cropped with 10% padding around the coarse mask and resampled to the size same as the input size of the model, inference is only performed once without the sliding window approach.

### 3.3   Ablation study

In the ablation study, as the baseline, fully supervised learning (FSL) is performed to train both coarse and fine segmentation models using only the labeled data. In other experiments, the coarse segmentation is fixed to the proposed method, and different training pipelines are used for the fine segmentation. First, FSL is applied to the fine segmentation using only the student model. Also, we conduct experiments with a single teacher and a student framework: FSL with KD, SSL with KD, SSL with pseudo-supervision, and SSL with KD and pseudo-supervision. We investigate the isolated effect of cross-teaching by training two cross-teaching students. Moreover, the proposed method's variants with no KD, no pseudo-supervision, and teachers sharing the same architecture, respectively, are performed. In all experiments, the network architectures of teacher and student models and training protocols are the same as in the proposed method. For experiments with two models of the same size for inference, we choose the one with transposed convolutions. Otherwise, the student model is used for inference.

# 4   Results and discussion

All DSC results for the experiments are obtained via the validation leaderboard of the MICCAI FLARE 2022 challenge. Also, detailed results, including efficiency analysis, are processed privately and provided by the challenge organizers based on submissions using Docker containers.

## 4.1   Ablation study

Table 3 shows the results of the ablation study. The baseline FSL shows a mean DSC of 0.7712, which slightly increases to 0.7812 when the proposed method is performed for the coarse segmentation. Applying KD to the basic FSL model yields an improved mean DSC of 0.8261 from 0.7812. This is better than the experiments on SSL with a single teacher and a student, which implies that ineffective use of unlabeled data only hinders the training of the student model. When unlabeled data is effectively exploited by the cross-teaching strategy, it shows better results than the FSL with KD even without the teacher model. Although there is little performance gain with pseudo-supervision from cross teaching teachers only, KD and combined use of KD and pseudo-supervision improve results. Moreover, teachers with slightly different decoders achieve better results than those with the same decoders, which is consistent with the results in Mutual Consistency Training  [34].

**Table 3.** Ablation study results on the MICCAI FLARE 2022 validation set. The baseline uses only the labeled data to train both coarse and fine segmentations, whereas, in the rest of the experiments, the proposed method is used for training the coarse segmentation and each row shows the training settings for the fine segmentation. CTS and CTT mean cross teaching between two students and two teachers, respectively. $CTT_{SD}$ uses teachers with decoders with the same architecture. *The one with transposed convolutions out of two models is used for inference.

| # of T | # of S | SSL | KD | PSV | Cross Teaching | Mean DSC |
|---|---|---|---|---|---|---|
| 0 | 1 | | | | | 0.7712±0.1193 (baseline) |
| 0 | 1 | | | | | 0.7812±0.1121 |
| 1 | 1 | | ✓ | | | 0.8261±0.1107 |
| 1 | 1 | ✓ | ✓ | | | 0.8227±0.1122 |
| 1 | 1 | ✓ | | ✓ | | 0.8234±0.1101 |
| 1 | 1 | ✓ | ✓ | ✓ | | 0.8173±0.1149 |
| 0 | 2 | ✓ | | | CTS | 0.8296±0.1092* |
| 2 | 1 | ✓ | | ✓ | CTT | 0.8297±0.1111 |
| 2 | 1 | ✓ | ✓ | | CTT | 0.8407±0.1075 |
| 2 | 1 | ✓ | ✓ | ✓ | $CTT_{SD}$ | 0.8394±0.1086 |
| 2 | 1 | ✓ | ✓ | ✓ | CTT | **0.8429±0.1043 (proposed)** |

### 4.2   Quantitative results on validation set

The proposed method shows a mean DSC of 0.8429±0.1043 and a mean NSD of 0.8990±0.0755 in the MICCAI FLARE 2022 validation set (Table 4). While large organs such as the liver or spleen are well segmented with DSC higher than 0.9, the proposed method works relatively poorly for adrenal glands and gallbladder. This may be attributed to the weakness of overlap-based metrics, including DSC, to small objects, since the proposed method depends on the Dice loss [31].

**Table 4.** Segmentation results on the MICCAI FLARE 2022 validation set.

| Organ | DSC | NSD |
|---|---|---|
| Liver | 0.9711±0.0214 | 0.9762±0.0406 |
| RK | 0.9095±0.2092 | 0.9177±0.2221 |
| LK | 0.8975±0.2163 | 0.9053±0.2240 |
| Spleen | 0.9593±0.0417 | 0.9704±0.0663 |
| Pancreas | 0.8575±0.0529 | 0.9468±0.0526 |
| Aorta | 0.9383±0.0249 | 0.9744±0.0544 |
| IVC | 0.8781±0.0963 | 0.8855±0.1254 |
| RAG | 0.6907±0.1533 | 0.8383±0.1684 |
| LAG | 0.6578±0.2009 | 0.7876±0.2228 |
| Gallbladder | 0.7165±0.3546 | 0.7225±0.3632 |
| Esophagus | 0.8189±0.1200 | 0.9199±0.1187 |
| Stomach | 0.8959±0.1647 | 0.9288±0.1563 |
| Duodenum | 0.7672±0.1281 | 0.9139±0.0872 |
| Mean | 0.8429±0.1043 | 0.8990±0.0755 |

### 4.3   Qualitative results on validation set

Figure 5 illustrates the example segmentation results of the baseline FSL model and the proposed method from the MICCAI FLARE 2022 validation set. Whereas baseline and proposed methods yield satisfactory results for routine contrast-enhanced CT images and healthy organs, the proposed method shows better results for CT with noise and non-portal contrast phases and lesion-affected organs. However, the proposed fails in some cases with large lesions or out-of-distribution diseases such as hiatal hernia or large amounts of ascites.

### 4.4   Segmentation efficiency results on validation set

The segmentation efficiency results are acquired in the private testing environment of the MICCAI FLARE 2022 challenge (Table 5). In the validation set, the mean running time of the proposed method is 28.89 s with a range of 24.77–48.43 s. The maximum GPU memory usage is 2025 MB for all cases. The areas under the GPU memory-time and CPU utilization-time curves shows a mean of

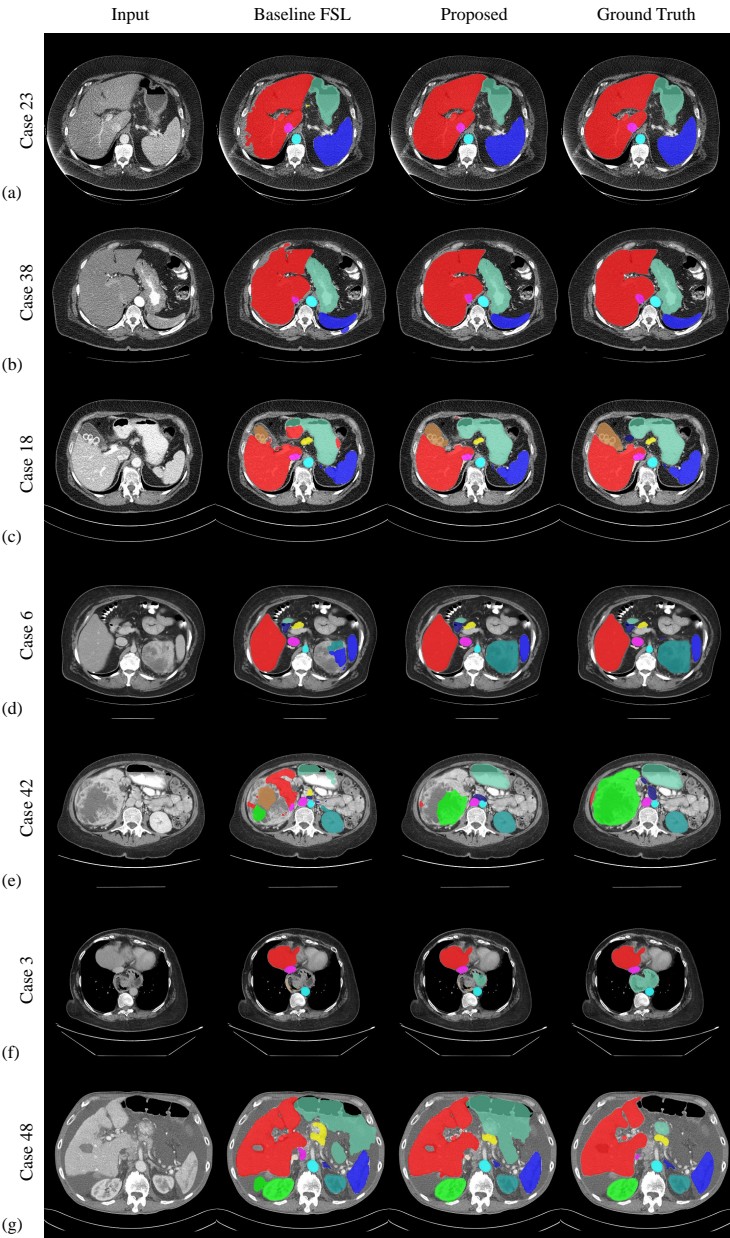

**Fig. 5.** Example cases from the MICCAI FLARE 2022 validation set. The first column is the CT image, the last column is the ground truth, and the second and third columns show the segmentation by the baseline fully supervised model and the proposed method, respectively. Descriptions for each row are as follows: (a) noisy image, (b) arterial phase contrast CT, (c) gallstones, (d) left kidney tumor, (e) large right kidney tumor, (f) hiatal hernia, and (g) large amounts of ascites.

27167 MB·s and 596.95 %·s, respectively, and a range of 25890–37132 MB·s and 466.76–990.31 %·s, respectively.

**Table 5.** Testing environments in MICCAI FLARE 2022 challenge.

| | |
|---|---|
| Windows/Ubuntu version Ubuntu 20.04 | |
| CPU | Intel® Xeon(R) W-2133 CPU @ 3.60GHz × 12 |
| RAM | 32G (Available memory 28G) |
| GPU | NVIDIA QUADRO RTX5000 (16G) |

### 4.5   Results on test set

The proposed method ranked 10th in the MICCAI FLARE 2022 test phase. The segmentation results showed a mean DSC of 0.8520±0.0987 and a mean NSD of 0.9137±0.0666 (Table 6). The mean running time was 28.16 s. The areas under the GPU memory-time and CPU utilization-time curves showed a mean of 23092 MB·s and 575 %·s, respectively.

**Table 6.** Segmentation results in the MICCAI FLARE 2022 test phase.

| Organ | DSC | NSD |
|---|---|---|
| Liver | 0.9763±0.0154 | 0.9859±0.0244 |
| RK | 0.9332±0.1672 | 0.9471±0.1742 |
| LK | 0.9420±0.1179 | 0.9542±0.1306 |
| Spleen | 0.9471±0.1386 | 0.9634±0.1445 |
| Pancreas | 0.8204±0.1002 | 0.9281±0.0961 |
| Aorta | 0.9375±0.0469 | 0.9748±0.0634 |
| IVC | 0.8850±0.0858 | 0.9050±0.0991 |
| RAG | 0.7338±0.1200 | 0.8808±0.1429 |
| LAG | 0.7135±0.1361 | 0.8559±0.1488 |
| Gallbladder | 0.7312±0.3493 | 0.7370±0.3567 |
| Esophagus | 0.7721±0.1398 | 0.8777±0.1554 |
| Stomach | 0.9254±0.0947 | 0.9569±0.0982 |
| Duodenum | 0.7587±0.1177 | 0.9109±0.1021 |
| Mean | 0.8520±0.0987 | 0.9137±0.0666 |

### 4.6   Limitations and future work

Although the idea of KD from SSL-based teachers can be applied to any kind of SSL design, this study only uses the cross teaching method, but there are other state-of-the-art SSL methods, including uncertainty-aware strategies [35,21]. Also, for KD, other losses than the KL divergence loss and other distillation methods

such as feature-based or relation-based KD can be utilized [7]. Moreover, we only use depthwise separable convolutions to build a student model, but other efficient building blocks such as the spatial pyramid module in ESPNet [25] may be a better choice. Comparisons of different SSL designs, KD methods, and efficient network architectures should be addressed in future work.

## 5    Conclusion

This study combines several methods for efficient semi-supervised abdominal organ segmentation in CT. A whole-volume-based coarse-to-fine framework and depthwise separable convolutions contribute to efficiency. Cross teaching and pseudo-supervision are applied to utilize unlabeled data. Also, models with slightly different decoders further enhance the effect of cross teaching. Finally, knowledge distillation enables the joint use of model compression and semi-supervised learning. The proposed method showed mean Dice scores of 0.8429 and 0.8520 in the MICCAI FLARE 2022 validation and test sets, respectively.

**Acknowledgements** The author of this paper declares that the segmentation method implemented for participation in the FLARE 2022 challenge has not used any pre-trained models or additional datasets other than those provided by the organizers. Also, the proposed solution is fully automatic without any manual intervention.

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
