# OpenReview forum: "Knowledge Distillation from Cross Teaching Teachers for Efficient Semi-Supervised Abdominal Organ Segmentation in CT"
_MICCAI.org/2022/Challenge/FLARE_

### Official Review · Reviewer_LmEm · 2022-09-16
**Knowledge Distillation from Cross Teaching Teachers for Efficient Semi-Supervised Abdominal Organ Segmentation in CT**

**Rating:** 7
**Confidence:** 3

**Review:**

Strengths: The proposed model achieves efficient semi-supervised learning and obtains a mean Dice score of 0.8429 in the validation set.
Weaknesses:
- The ablation study results (training the proposed model with only 50 labeled cases) are not presented in the paper.
- Why do you choose patch-wise intensity normalization? Do you implement the patch-wise intensity normalization during training?
- The description of training protocols is not clear. Do you use all unlabeled data for network training? How do you choose the best model weight? Do you train the two teacher models and the student model simultaneously? How do you achieve coarse-to-fine segmentation during network training?
- It is better to add more elaborate figure captions in Figure 1 and Figure 2.

---

> ### Author Response · Authors · 2022-10-14
> **Thank you for the comments.**
>
> * The ablation study results (training the proposed model with only 50 labeled cases) are not presented in the paper\
> -> They are presented as the "baseline FSL" in the paper.
> * Why do you choose patch-wise intensity normalization? Do you implement the patch-wise intensity normalization during training?\
> -> Yes, patchwise normalization is performed during training. Whole-volume-wise normalization worked poorly for large field-of-view images or whole-body CTs.
> * The description of training protocols is not clear. Do you use all unlabeled data for network training? How do you choose the best model weight? Do you train the two teacher models and the student model simultaneously? How do you achieve coarse-to-fine segmentation during network training?\
> -> The section for the training protocols has been modified. All unlabeled data are used for training, but each epoch includes a random subset of the unlabeled data. The model weight at the last epoch (1000) is used. Each stage, the coarse and fine segmentation, of the framework is trained sequentially and consists of teacher and student models which are trained simultaneously.
> * It is better to add more elaborate figure captions in Figure 1 and Figure 2.\
> -> The captions in Figures 1 and 2 have been modified.

---

### Official Review · Reviewer_si11 · 2022-09-17
**MICCAI-FLARE**

**Rating:** 9
**Confidence:** 4

**Review:**

Pros:
1. Proposing a coarse-to-fine framework with two teacher models and a student model that combines knowledge distillation and cross teaching, a consistency regularization based on pseudo-labels, for efficient semi-supervised learning.
2. Do more work for improving DSC and NSC
3. knowledge distillation and cross teaching is strong perfomance.

Cons: Lack of analysis of algorithm acceleration time.

---

> ### Author Response · Authors · 2022-10-14
> **Thank you for the comments.**
>
> * Lack of analysis of algorithm acceleration time.\
> -> I agree the paper is focusing more on the semi-supervised learning than the efficiency. The coarse-to-fine structure is the main component for efficiency and the following description is added and modified in the methods:\
> *Such a two-stage framework lowers computation costs, especially
> in terms of memory use and running time, compared to the sliding window approach, which is a more common solution in medical image segmentation. Empirically, a single-stage segmentation led to poor segmentation results and long inference time in large field-of-view or whole-body CT images.*

---

### Official Review · Reviewer_DTqs · 2022-09-19
**Good paper, interesting ideas**

**Rating:** 6
**Confidence:** 3

**Review:**

+ Nice visualization
+ A clear and concise description of the method
+ Interesting idea
+ Strong improvement in comparison to measured baseline score

But, the baseline score itself is somewhat small, such that your improved score is same or smaller than the baseline scores of some other papers, which is as high as around .84-.86 DSC.

---

> ### Author Response · Authors · 2022-10-14
> **Thank you for the comments.**
>
> * But, the baseline score itself is somewhat small, such that your improved score is same or smaller than the baseline scores of some other papers, which is as high as around .84-.86 DSC.\
> -> The smaller baseline and improved scores are mainly due to the smaller patch size, compared to other papers in the FLARE challenge. To ensure the GPU memory within the 2GB tolerance of the challenge, hard resizing (whole CT volume to 96\*96\*96) and a small patch size were used, despite of compromising some segmentation performance.

---

### Meta-Review · Program_Chairs · 2022-09-28

**Recommendation:** Minor Revision
**Confidence:** 5

**Metareview:**

Nice paper.
Please address the reviewers' comments in the revised manuscript.